# Land Use and Land Cover Changes in Kabul, Afghanistan Focusing on the Drivers Impacting Urban Dynamics during Five Decades 1973–2020

Hayatullah Hekmat [1,2], Tauseef Ahmad [3], Suraj Kumar Singh [4,*], Shruti Kanga [5], Gowhar Meraj [6] and Pankaj Kumar [7]

1    ArMehr Bussiness Center, 2nd Floor B3, Tank Tell Onchi, Kabul 1006, Afghanistan; hayatullah.hs@gmail.com
2    Centre for Climate Change and Water Research, Suresh Gyan Vihar University, Jaipur 302017, India
3    Geological Engineering, Canakkale Onsekiz Mart University, 17100 Canakkale, Turkey; tauseefahmad@ogr.comu.edu.tr
4    Centre for Sustainable Development, Suresh Gyan Vihar University, Jaipur 302017, India
5    Department of Geography, School of Environment and Earth Sciences, Central University of Punjab, Bathinda 151401, India; shruti.kanga@cup.edu.in
6    Department of Ecosystem Studies, Graduate School of Agricultural and Life Sciences, The University of Tokyo, 1-1-1 Yayoi, Tokyo 113-8654, Japan; gowharmeraj@g.ecc.u-tokyo.ac.jp
7    Institute for Global Environmental Strategies, Hayama 240-0115, Japan; kumar@iges.or.jp
*    Correspondence: suraj.kumar@mygyanvihar.com

**Abstract:** This study delves into the patterns of urban expansion in Kabul, using Landsat and Sentinel satellite imagery as primary tools for analysis. We classified land use and land cover (LULC) into five distinct categories: water bodies, vegetation, barren land, barren rocky terrain, and buildings. The necessary data processing and analysis was conducted using ERDAS Imagine v.2015 and ArcGIS 10.8 software. Our main objective was to scrutinize changes in LULC across five discrete decades. Additionally, we traced the long-term evolution of built-up areas in Kabul from 1973 to 2020. The classified satellite images revealed significant changes across all categories. For instance, the area of built-up land reduced from 29.91% in 2013 to 23.84% in 2020, while barren land saw a decrease from 33.3% to 28.4% over the same period. Conversely, the proportion of barren rocky terrain exhibited an increase from 22.89% in 2013 to 29.97% in 2020. Minor yet notable shifts were observed in the categories of water bodies and vegetated land use. The percentage of water bodies shrank from 2.51% in 2003 to 1.30% in 2013, and the extent of vegetated land use showed a decline from 13.61% in 2003 to 12.6% in 2013. Our study unveiled evolving land use patterns over time, with specific periods recording an increase in barren land and a slight rise in vegetated areas. These findings underscored the dynamic transformation of Kabul's urban landscape over the years, with significant implications for urban planning and sustainability.

**Keywords:** land use/land cover; supervised classification; urban sprawl; Kabul; geoinformatics

## 1. Introduction

Land use/land cover change (LULCC) is a global process with significant implications for ecology [1]. Rapid alterations in LULCC primarily driven by urban sprawl and deforestation engender profound environmental impacts [2,3]. The prime concern associated with LULCC is the loss of biodiversity which, in turn, influences various climate change parameters, exacerbates surface runoff and urban heat, and contributes to groundwater table depletion [4–8]. In many developing countries, LULCC has intensified in response to burgeoning food demand, leading to prevalent practices such as deforestation and the expansion of agricultural areas [5,9–11]. The increase in LULCC over the past four decades has escalated the biodiversity crisis, interrupting ecological processes indispensable for

human survival [12–15]. These long-term changes catalyze climate change and act as precursors to various natural hazards [16,17].

Long-term urban studies have emerged as critical components for understanding LULCC. These studies evaluate the growth direction, intensity, and influencing factors of LULCC, as well as their impact on local, regional, and global parameters. Consequently, this provides valuable insights for land use management and urban sustainability planning [18]. The steep increase in population, with the majority dwelling in urban areas, has emerged as a significant driver of LULCC. Current trends suggest that nearly two-thirds of the world's population will inhabit urban areas by 2050, with the total population exceeding 11.2 billion by 2100 [4,19,20]. Previous research on LULC unveiled a considerable expansion of agricultural land, averaging a loss of 15 million hectares of forest land annually [1]. This transformation has posed significant challenges, particularly in Eastern and Southern Africa, due to high population demand [21–24]. Additionally, forest depletion resulting from human activities has been reported in various countries, including the United States, Nepal, India, and Mexico [25–27].

Remote sensing in conjunction with GIS has been extensively used to cost-effectively monitor the spatial and temporal dynamics of LULCC worldwide [28–31]. Satellite images are processed in remote sensing to achieve LULC classification, with supervised classification methods deploying various types of classifiers like k-nearest neighbor (kNN), random forest (RF), support vector machine (SVM), and maximum likelihood classifier (MLC) being most commonly used [32,33]. Afghanistan, a critical country in the Indian subcontinent linked with the Hindu Kush Himalayas, is ranked 9th in the world in the Fragile State Index 2021 [34]. The accelerated population growth in Kabul, Afghanistan's principal urban economic and commercial center, has expanded the city's coverage area. Accordingly, this study sought to comprehend the extent of LULCC in Kabul, discussing long-term changes in terms of directional and density growth patterns. We spatially represented and discussed the major causes responsible for the observed changes in the city.

## 2. Materials and Methods

### 2.1. Study Area

Kabul, the capital of Afghanistan, is the focal point of this study. This landlocked region encompasses a total area of 1030 km$^2$ (Figure 1). It is situated in the eastern segment of the country, with coordinates ranging from 34°17′ N to 34°41′ N latitude and 68°50′ E to 69°30′ E longitude. The city's elevation is approximately 1790 m above sea level. Kabul's geographic features are notable, surrounded by majestic mountains on most sides: Koh-e Shirdarwaza Mountain is positioned in the northeast, Koh-e Qrough Mountain in the southwest, and Koh-e Paghman Mountain in the east. The city is also flanked by the Kabul River, with its basin boasting a total catchment area of 76,908 km$^2$ [35]. Afghanistan, characterized by its rugged terrain, is dominated by the Hindu Kush Mountains that run from the northeast to the southwest, bisecting the country into three distinct regions. Two-thirds of the area comprise the Central Highlands, an extension of the Himalayas. The southern plateau forms about a quarter of the country's land, while the remaining terrain, the Northern Plains, is the most fertile.

Climate patterns in Kabul fluctuate considerably throughout the year and have Dfb-warm summer continental climate (Köppen 1936; Rubel and Kottek 2010). Summers (June–July–August) present mean temperatures ranging from 24 °C to 33 °C, while winters (December–January–February) record temperatures between 0 °C and 8 °C. July is the warmest month and January is the chilliest [36]. In certain extreme weather conditions, temperatures can plummet to −45 °C or escalate beyond 50 °C in cities like Ferozkoh and Zaranj. Precipitation patterns exhibit similar variability, with the country receiving an average annual precipitation of 300 mm. Higher regions witness precipitation from November to April due to winter storms originating from the Mediterranean region [37,38].

Kabul holds the distinction of being the fifth fastest-growing city globally, encompassing 14 administrative and 22 municipal districts, colloquially referred to as Nahiyes [39].

As the primary urban hub of Afghanistan, Kabul serves as the crucible for the country's economic, cultural, and political advancement. However, its rapid urbanization, juxtaposed with the local authorities' limited capacity to manage this growth effectively, has resulted in an escalation of impoverished and vulnerable populations, posing significant challenges for urban planning and management [19]. Despite the burgeoning urbanization, the city's infrastructural growth has struggled to keep pace with the demands of its swiftly expanding population. This is particularly stark when one considers that Kabul was originally designed to accommodate a populace of merely 0.7 million. This incongruity has resulted in approximately 70 percent of Kabul's inhabitants residing in informal areas—zones that have grown organically without formal urban planning [40].

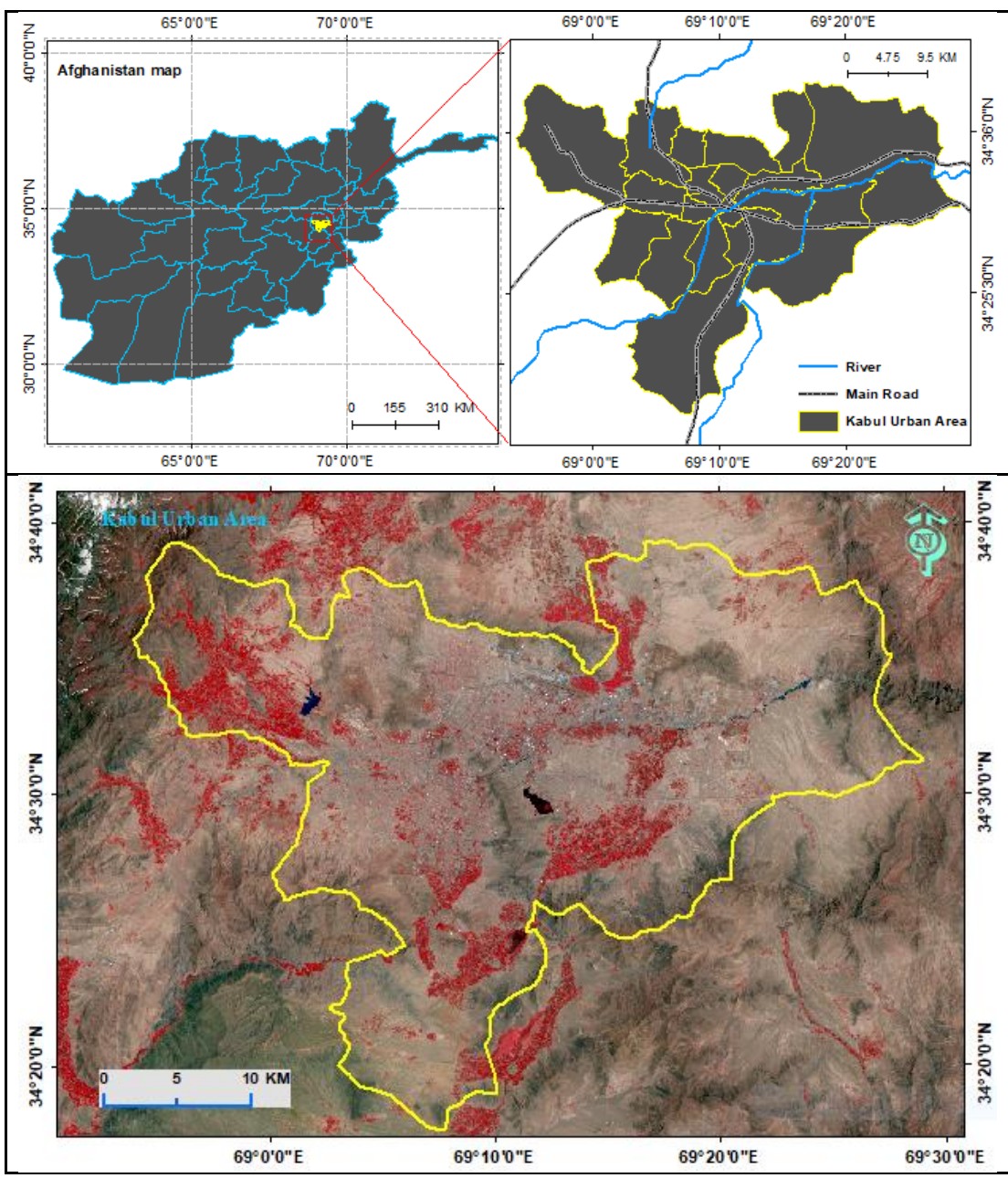

**Figure 1.** Kabul represented as the study area using a Sentinel 2 satellite image during 12 July 2020.

*2.2. Datasets and Methods Involved*

This study includes the use of multi-temporal satellite images to understand the land use/land cover changes in Kabul, Afghanistan. The Landsat and Sentinel satellite images

from 1973 to 2013 were obtained from the USGS (United States Geological Survey) and the Global Visualization (GloVis) portal for the period of 1973 to 2013, whereas the Sentinel data for the year 2020 was obtained from European Space Agency (https://www.esa.int/) (accessed on 31 September 2020) (Table 1). A total of 7 satellite images of the summer season (June–September) of each year were acquired. The data were preprocessed before further analysis which includes geometric registration, and radiometric and atmospheric corrections to avoid spurious results produced from these issues [41]. The study area of the city of Kabul was clipped out from all the satellite images using Erdas Imagine v.2015 software using shapefile prepared from the topographic map of Afghanistan (USGS). Software like ENVY 5.3 and Arc GIS v.10.7 used for further analysis and preparation of maps.

**Table 1.** Details of data used in the methodology.

| Date | Satellite/Sensor | Resolution/Scale (Meters) | Source |
|---|---|---|---|
| 10 September 1973 | Landsat 1 MSS | 60 | https://www.usgs.gov (accessed on 31 September 2020) |
| 7 March 1983 | Landsat 4 | 60 | https://www.usgs.gov (accessed on 31 September 2020) |
| 8 June 1993 | Landsat 5 | 30 | https://www.usgs.gov (accessed on 31 September 2020) |
| 11 May 2003 | Landsat 7 | 30 | https://www.usgs.gov (accessed on 31 September 2020) |
| 15 June 2013 | Landsat 8/ OLI | 30 | https://www.usgs.gov (accessed on 15 January 2021) |
| 12 July 2020 | MSI | 10 | https://www.esa.int/ (accessed on 31 September 2020) |
| 2005 | GIS Shape file | 1:250,000 | USGS |

We employed the maximum likelihood algorithm in the supervised classification method to generate land use and land cover (LULC) maps for the years ranging from 1973 to 2020 [42]. To fine-tune the training samples for recognized land-cover types, a variety of image enhancement techniques were applied. These include contrast stretching, histogram equalization, spatial filtering, and pan-sharpening [41]. These techniques were guided by the available ground truth data, particularly for the 2020 dataset. In this most recent dataset, 100 well-distributed training samples were selected, covering five distinct LULC classes: vegetation, built-up, water bodies, barren land, and barren rock. For historical LULC classifications corresponding to 1973, 1983, 1993, 2003, and 2013, we selected optimal training samples based on the highest achievable accuracy, constrained by the limited availability of reliable ground truth data for those years. For these historical datasets, spectral signatures from similar classes in different years were compared for internal consistency, serving as an indirect form of validation to be used for training samples. For the 1973 and 1983 images with a different spatial resolution (60 m), we explicitly state that the accuracy metrics may not be directly comparable to later years with higher resolutions due to the varied spectral signatures for these years.

For validation, we adopted a multi-faceted approach to account for the varying quality of available data across the years. A consistent set of 100 well-distributed sampling sites were used for accuracy assessments for all years. These sites were carefully selected to be representative of the five distinct LULC classes. For the 2020 dataset, ground-truth accuracy assessments were conducted at these 100 sites, both pre- and post-classification, aiding in the calculation of the Kappa coefficient [43]. In the case of the 2013 dataset, the accuracy of the classification was cross-verified using Google Earth historical imagery. For other historical years, where high-quality imagery was lacking, we used indirect methods

for validation. For the 1973 and 1983 datasets, which are characterized by a coarser 60 m spatial resolution, we used ancillary historical archived maps as an alternative means of validation. Although this approach is not as rigorous as ground-truthing, it provides a reasonable measure of accuracy, especially when considered alongside the high Kappa coefficients and overall accuracy percentages observed. Moreover, we also employed change detection algorithms to scrutinize abrupt or unrealistic changes in LULC classes between the years, further strengthening the internal validity of the study. Despite the limitations and variations in spatial resolution and data quality, these validation methods collectively enhance the internal consistency and credibility of our LULC assessments [44–46]. The overall accuracy and Kappa coefficient values for the years 1973 to 2020, as presented in Table 2, serving as statistical evidence of this consistency.

**Table 2.** Accuracy assessment of classified images during 1973 to 2020.

| LULC Classification/Years | Overall Accuracy (%) | Kappa Coefficient |
|---|---|---|
| 10 September 1973 | 88.00 | 0.525 |
| 7 March 1983 | 89.33 | 0.613 |
| 8 June 1993 | 90.00 | 0.631 |
| 11 May 2003 | 93.33 | 0.758 |
| 15 June 2013 | 93.33 | 0.813 |
| 12 July 2020 | 96.67 | 0.964 |

An illustrative overview of the comprehensive methodology employed is provided in Figure 2.

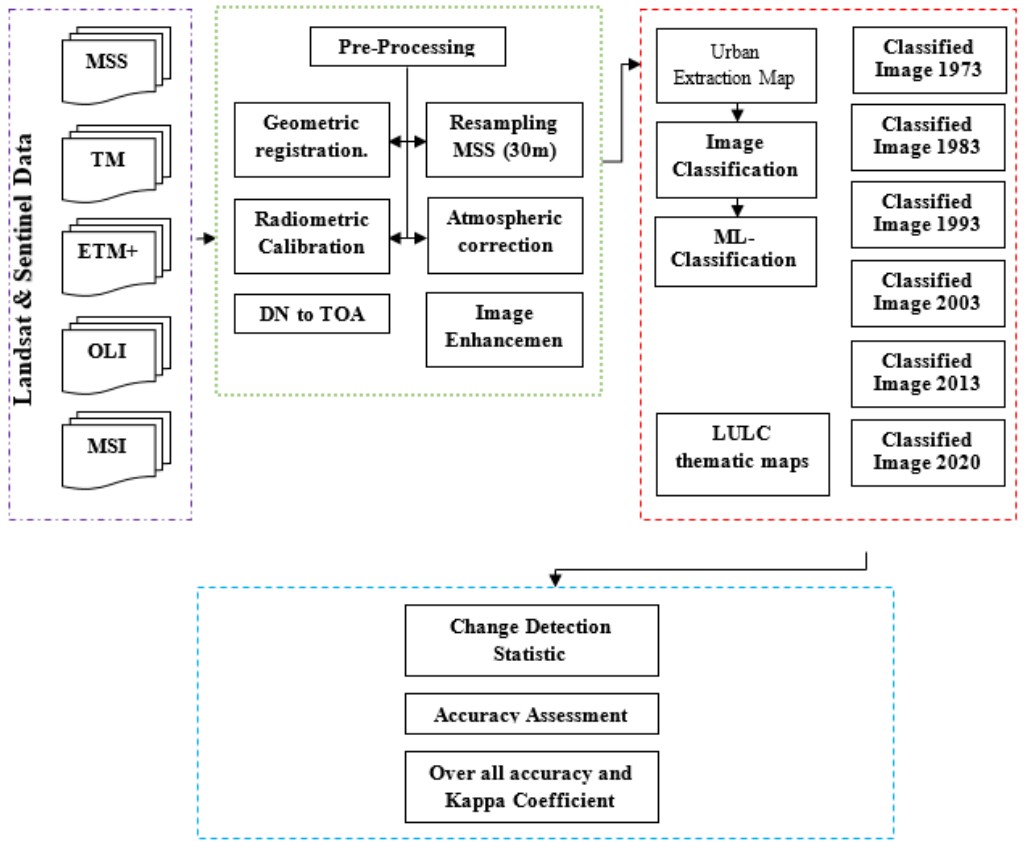

**Figure 2.** The flow chart of methodology adopted in this research.

The Kappa coefficient (k) is mathematically expressed as:

$$k = \left\{ N\sum_{i=1}^{r}(X_{ii}) - N\sum_{i=1}^{r}(X_{i+}.X_{+i}) \right\} / N^2 - \sum_{i=1}^{r}(X_{i+}.X_{+i})$$

where, r represents the number of rows in the error matrix; $X_{ii}$ represents the number of observations in row i and column i; $X_{i+}$ is the total of observations in row i; $X_{+i}$ is the total of observations in the column i; N is the total number of observations included in the matrix.

## 3. Results and Discussion

In this study, long-term built-up growth in the city of Kabul from 1973–2020 was analyzed. Kabul has faced several changes in recent decades which can be monitored through satellite image classified maps.

### 3.1. Land Use and Land Cover Changes during 1973–1983

A comprehensive analysis of land use change in the Kabul Urban Area (KUA) between 1973 and 1983 revealed significant shifts in vegetated land use, barren land use, built-up areas, and barren rocky terrain (Figures 3 and 4). The land use maps from these two years provide a visual representation of these changes. In 1973, the built-up area, signifying developed or urbanized land, constituted 21.39% of the KUA's total area. By 1983, this percentage had risen to 26.07%, indicating a noticeable urban expansion over the decade (Table 3). This upsurge could be attributed to a variety of factors, including population growth, economic development, and changes in land use policy. Agricultural land use also saw an increase in 1983, largely due to regions that were not accounted for in the 1973 data. These previously unrecorded areas were classified as agricultural in the subsequent land use map. A stark transformation was observed in the proportion of barren land, decreasing from 44.35% in 1973 to 35.53% in 1983. This reduction could be a result of land conversion to accommodate expanding built-up and agricultural areas [47]. The water bodies also showed a slight increase from 1.76% in 1973 to 2.29% in 1983. This augmentation might be associated with natural water accumulation or human-led water management initiatives. Lastly, the land classified as barren rocky demonstrated a decrease from 26.29% in 1973 to 23.45% in 1983. This reduction suggests that these rocky terrain areas were possibly repurposed for agricultural or built-up use, marking another significant evolution in the city's land use dynamics [48].

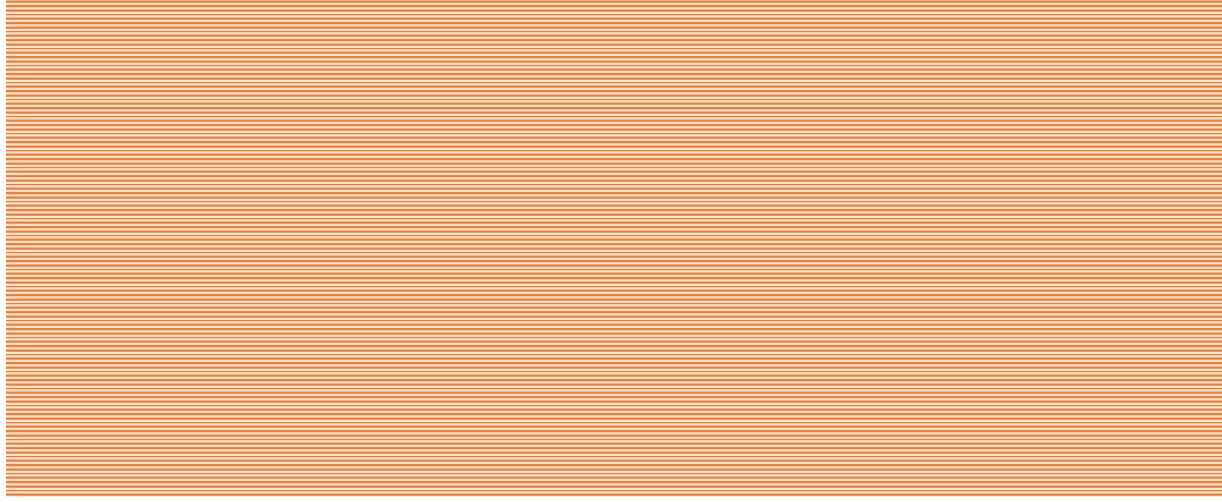

**Figure 3.** Landsat Satellite Images of the Kabul Urban Area in 1973 and 1983.

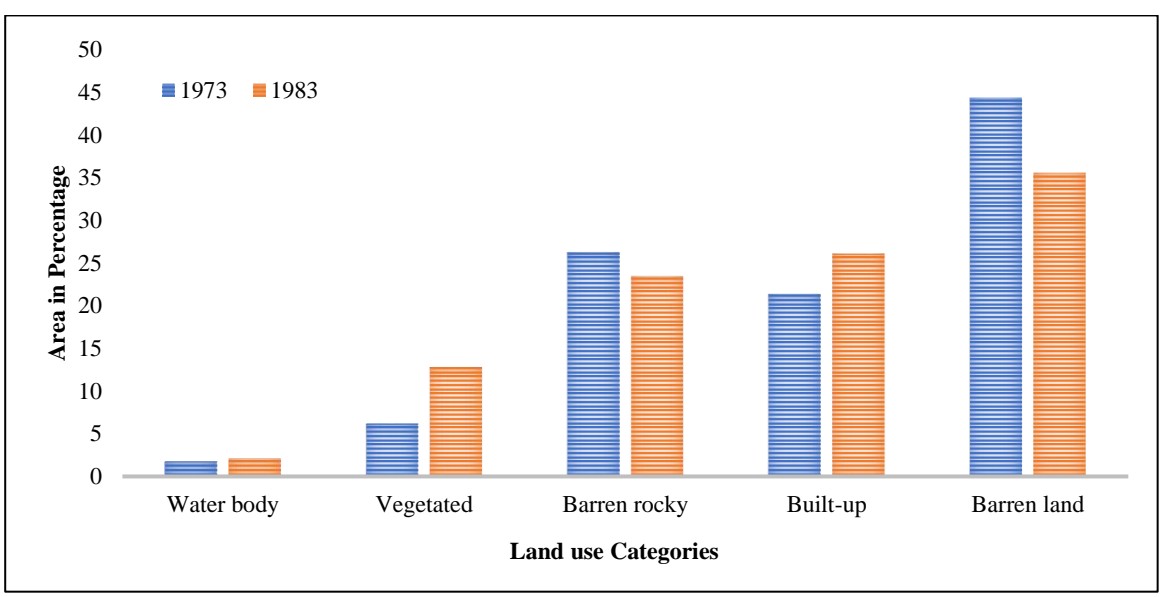

**Figure 4.** Graphical representation of land use land cover classification during 1973–1983.

**Table 3.** Land use/land cover classification of satellite images on 10 September 1973 and 7 March 1983.

| Year | Vegetation | | Built-up | | Water Bodies | | Barren Land | | Barren Rocky | |
|---|---|---|---|---|---|---|---|---|---|---|
| | Km$^2$ | (%) | Km$^2$ | (%) | Km$^2$ | (%) | Km$^2$ | (%) | Km$^2$ | (%) |
| 1973 | 64.0 | 6.21 | 220.3 | 21.39 | 18.1 | 1.76 | 456.8 | 44.35 | 270.8 | 26.29 |
| 1983 | 132.5 | 12.86 | 268.5 | 26.07 | 23.6 | 2.29 | 363.9 | 35.33 | 241.5 | 23.45 |

*3.2. Land Use and Land Cover Changes during 1983–1993*

An in-depth analysis of land use and land cover changes between 1983 and 1993, utilizing Landsat 4 and Landsat 5 satellite imagery, presented intriguing results for the Kabul Urban Area (KUA) [49–51]. Notably, the built-up and barren land use categories underwent major transformations during this period. In 1983, the built-up area, signifying urbanization, accounted for 26.07% of KUA (Figures 5 and 6). By 1993, however, this percentage had decreased to 22.32% (Table 4). This reduction is somewhat unusual for urban areas, which generally expand over time. This regression in the built-up area might be attributed to the civil unrest that plagued Afghanistan during this period [52,53]. The civil conflict could have dissuaded people from living in Kabul, resulting in a decrease in developed land. Simultaneously, the decade saw considerable mismanagement of surface water, causing its representation to reduce from 2.29% in 1983 to a mere 0.5% in 1993 [54]. This dramatic decrease signifies potential water scarcity or poor water management practices, further aggravated by the ongoing civil war. On the other hand, barren land saw an increase from 35.53% in 1983 to 38.86% in 1993. This might reflect a pattern of desertification or diminished productivity, potentially linked to the reduced built-up and water body areas [55]. Additionally, a slight increase was recorded in the vegetated area, from 12.86% in 1983 to 13.98% in 1993. Despite the overall volatile situation, this increase suggests some successful efforts toward maintaining or enhancing green cover in the KUA during the decade [56].

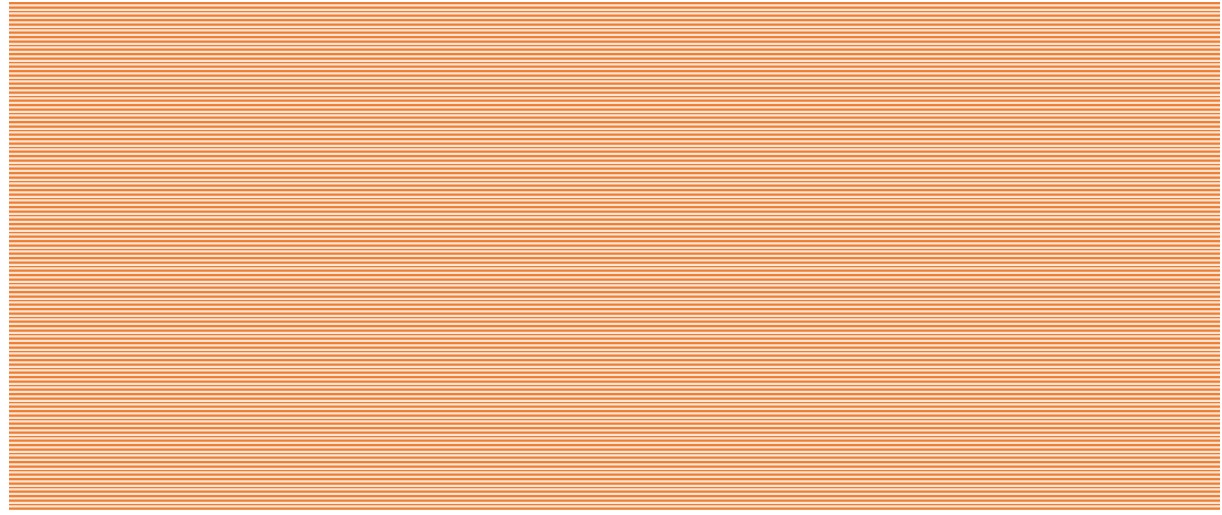

**Figure 5.** Landsat Satellite Images of the Kabul Urban Area in 1983 and 1993.

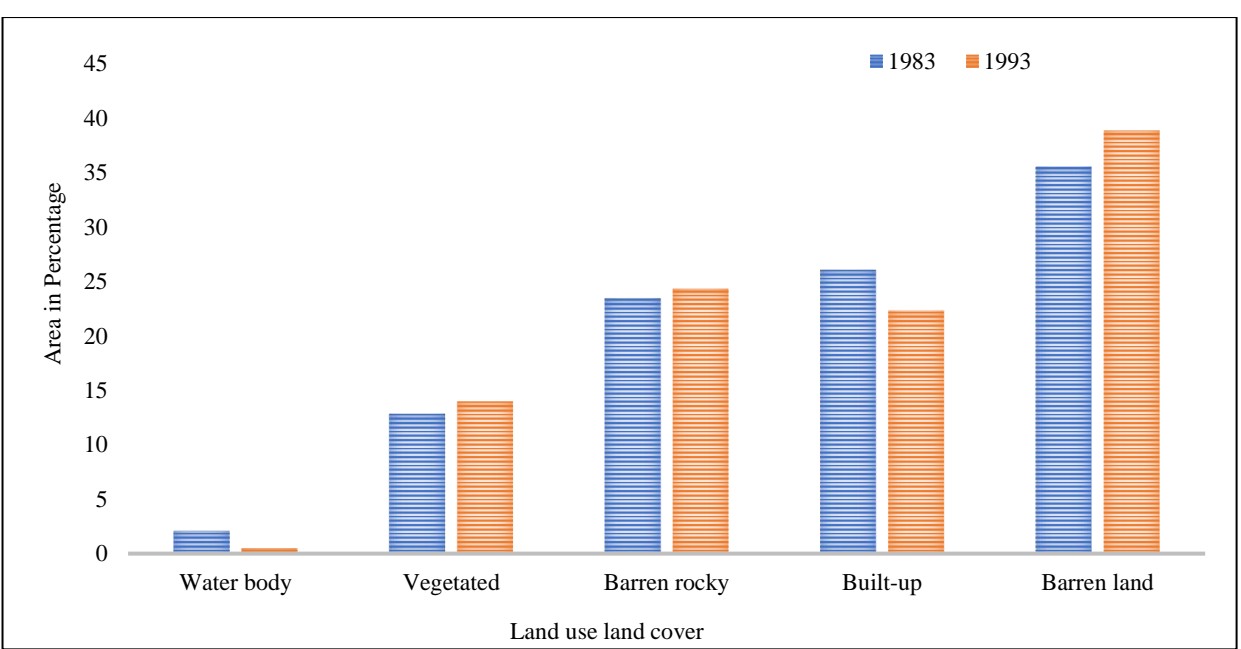

**Figure 6.** Graphical representation of land use/land cover classification during 1983–1993.

**Table 4.** Land use/land cover classification of satellite images on 7 March 1983 and 8 June 1993.

| Year | Vegetation | | Built-up | | Water Bodies | | Barren Land | | Barren Rocky | |
|------|------------|------|----------|------|--------------|------|-------------|------|--------------|------|
| | Km² | (%) | Km² | (%) | Km² | (%) | Km² | (%) | Km² | (%) |
| **1983** | 132.5 | 12.86 | 268.5 | 26.07 | 23.6 | 2.29 | 363.9 | 35.33 | 241.5 | 23.45 |
| **1993** | 144.0 | 13.98 | 229.9 | 22.32 | 5.6 | 0.5 | 400.4 | 38.86 | 250.6 | 24.33 |

### 3.3. Land Use and Land Cover Changes during 1993–2003

A comprehensive study was conducted on the land use changes in the Kabul Urban Area (KUA) over a decade, using satellite data from Landsat 5 (1993) and Landsat 7 (2003). The analysis revealed remarkably stable land use patterns between 1993 and 2003, with minor fluctuations in certain categories. One plausible explanation for the relative stability in land use during this decade might be the continued impact of war and internal insecurity in Afghanistan. These conditions might have hindered significant urban development or

rural land transformation, resulting in relatively consistent land use categories over the decade [57]. Despite the overall consistency, a minor change was observed in the area of surface water. From accounting for just 0.5% of the KUA's total area in 1993, surface water representation increased to 2.51% in 2003 (Figures 7 and 8). This increase could reflect improvements in water management or natural accumulation due to precipitation changes during the period (Table 5). Simultaneously, a slight decrease was recorded in the built-up area from 22.32% in 1993 to 20.40% in 2003. This reduction, although small, indicates a minor decline in urbanization over the decade, possibly due to continued sociopolitical challenges affecting the region's development and settlement patterns [58].

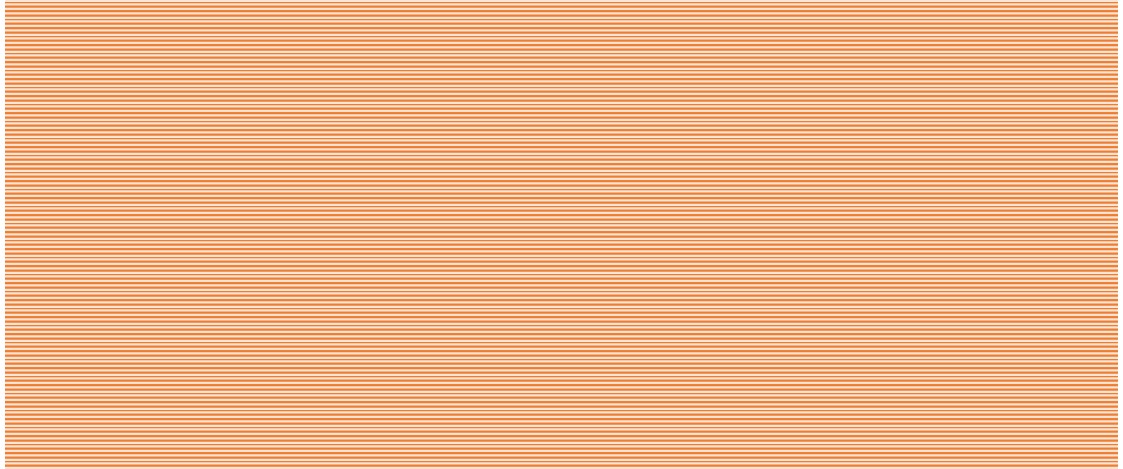

**Figure 7.** Landsat Satellite Images of Kabul Urban Area 1993 and 2003.

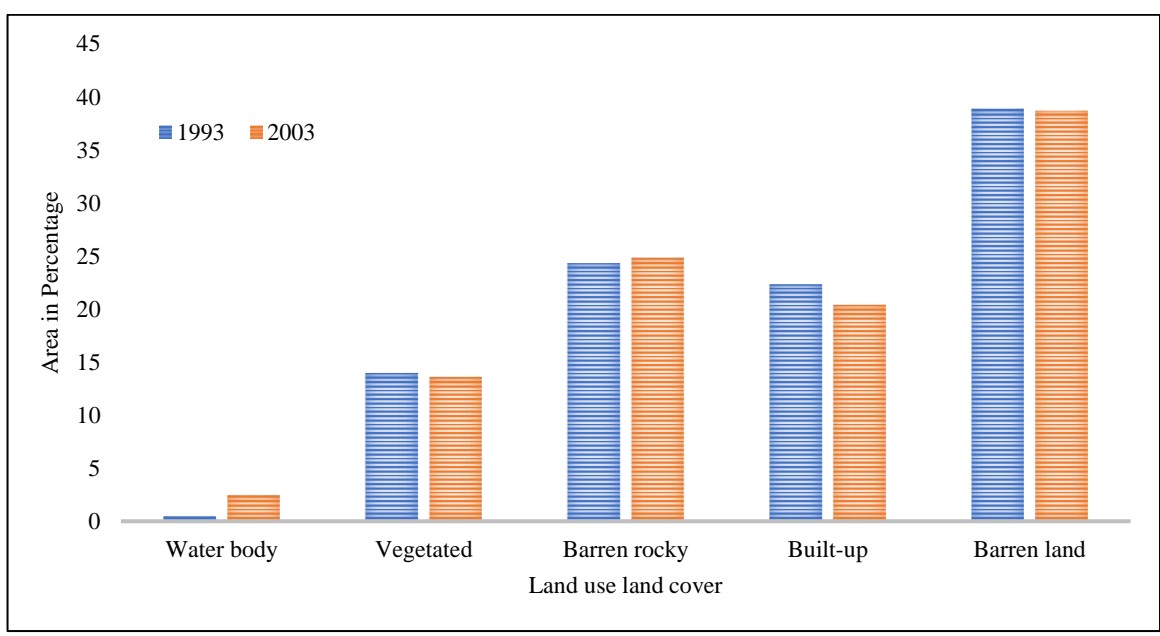

**Figure 8.** Graphical representation of land use land cover classification during 1993–2003.

**Table 5.** Land use/land cover classification of satellite images during 8 June 1993 and 11 May 2003.

| Year | Vegetation | | Built-Up | | Water Bodies | | Barren Land | | Barren Rocky | |
|---|---|---|---|---|---|---|---|---|---|---|
| | Km² | (%) | Km² | (%) | Km² | (%) | Km² | (%) | Km² | (%) |
| 1993 | 144.0 | 13.98 | 229.9 | 22.32 | 5.6 | 0.5 | 400.4 | 38.86 | 250.6 | 24.33 |
| 2003 | 140.2 | 13.61 | 210.1 | 20.4 | 25.9 | 2.51 | 398.2 | 38.66 | 255.6 | 24.82 |

### 3.4. Land Use Land Cover Changes during 1993–2003

The analysis of Landsat 7 (2003) and Landsat 8 (OLI—TIRS) (2013) satellite data unveiled substantial land use changes within the Kabul Urban Area (KUA) over the course of a decade [58]. The most pronounced shift was observed in the built-up land category, which swelled from 20.4% of the city's total area in 2003 to 29.91% in 2013 (Figures 9 and 10). This significant increase in built-up land is likely reflective of the return and resettlement of Afghan citizens following the tumultuous periods of conflict. However, while urban expansion was occurring, shifts were also noted in natural land features. Specifically, there was a noticeable reduction in the extent of water bodies and vegetated lands. The coverage of water bodies dipped from 2.51% in 2003 to 1.30% in 2013 (Table 6). This decrease could be attributed to the increased built-up area, which may have encroached on water bodies, or possibly due to changes in precipitation or water management practices. Additionally, the extent of vegetated land showed a slight contraction, falling from 13.61% in 2003 to 12.6% in 2013. This trend might be linked to the urban expansion, as natural vegetation areas could have been replaced with built-up environments [58]. Alternatively, it might reflect changes in agricultural practices or natural cycles of vegetation growth and decline. The simultaneous expansion of built-up areas and contraction of natural land features highlight the inherent tension between urban development and environmental conservation [59].

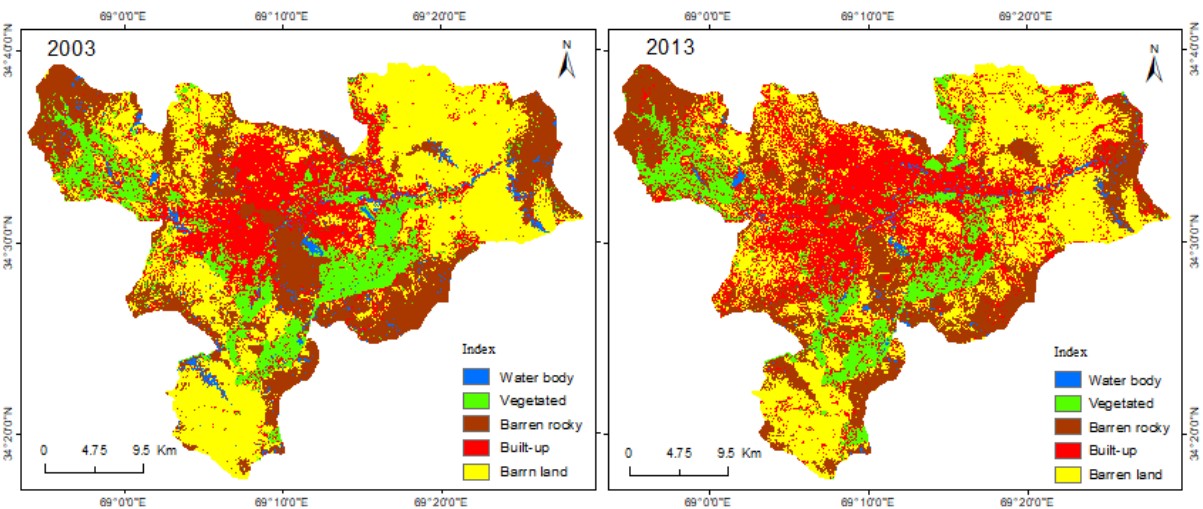

**Figure 9.** Landsat Satellite Images of the Kabul Urban Area in 2003 and 2013.

**Table 6.** Land use/land cover classification of satellite images during 11 May 2003 and 15 June 2013.

| Year | Vegetation | | Built-up | | Water Bodies | | Barren Land | | Barren Rocky | | Total |
|------|------|-----|------|-----|------|-----|------|-----|------|-----|-----|
| | Km² | (%) | Km² | (%) | Km² | (%) | Km² | (%) | Km² | (%) | (%) |
| 2003 | 140.2 | 13.61 | 210.1 | 20.4 | 25.9 | 2.51 | 398.2 | 38.66 | 255.6 | 24.82 | 100.00 |
| 2013 | 126.7 | 12.3 | 308.07 | 29.91 | 16.5 | 1.60 | 343.0 | 33.3 | 235.8 | 22.89 | 100.00 |

### 3.5. Land Use and Land Cover Changes during 2013–2020

The comparison of Landsat 8 (2013) and Sentinel 2A (2020) satellite data for the Kabul Urban Area (KUA) reveals significant shifts in land use and land cover categories over the seven-year period, most notably in barren land, built-up areas, and barren rocky terrain. In stark contrast to the expansion seen between 2003 and 2013, built-up land experienced a decline from 29.91% in 2013 to 23.84% by 2020 (Figures 11 and 12). This contraction could be due to a variety of factors including changes in urban planning policies, economic conditions, or migration patterns. Understanding the drivers of this trend would require further investigation. The percentage of barren land also showed a decrease, dropping from 33.3% in 2013 to 28.4% in 2020 (Table 7). This reduction could suggest a shift in land

use practices or the presence of significant land reclamation projects. On the other hand, barren rocky terrain exhibited a marked increase, swelling from 22.89% in 2013 to 29.97% in 2020. This uptick might indicate an increase in land degradation or it could be a result of shifts in the city's topography due to human activities or natural processes. Overall, these significant alterations in land use and cover demonstrate the dynamic nature of urban landscapes and underline the importance of continual monitoring for effective urban planning and environmental management [60,61].

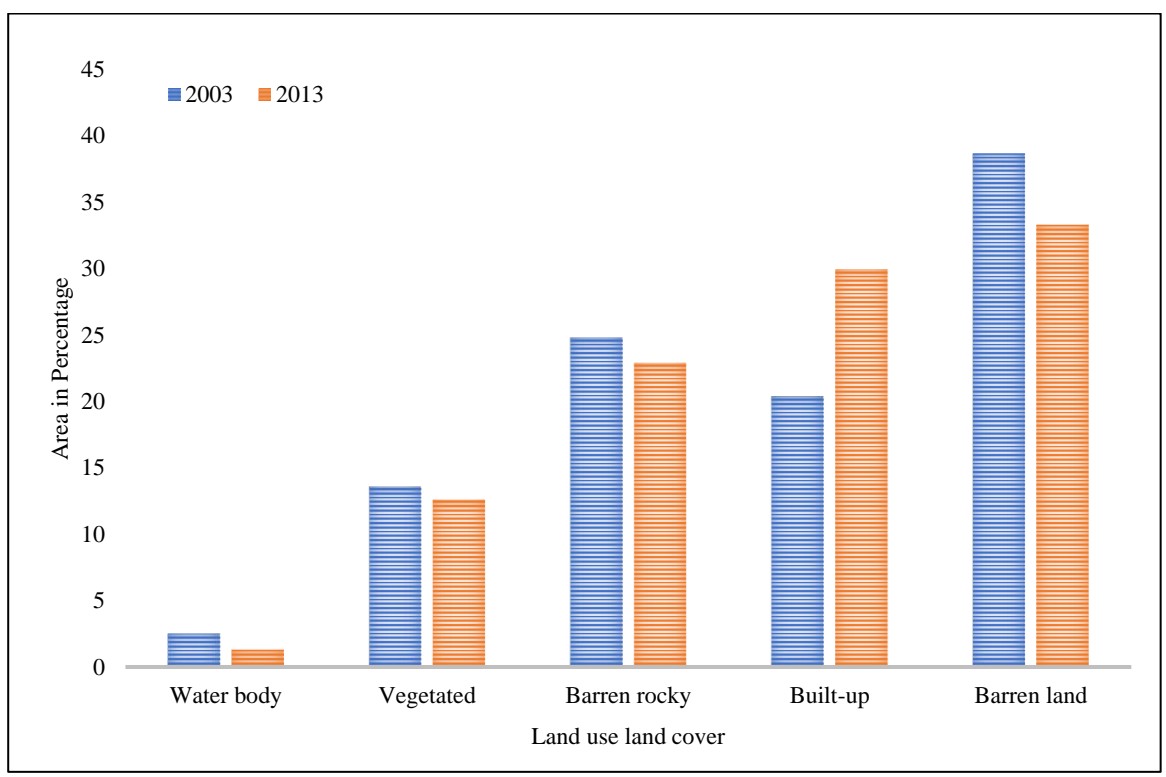

**Figure 10.** Graphical representation of land use/land cover classification during 2003–2013.

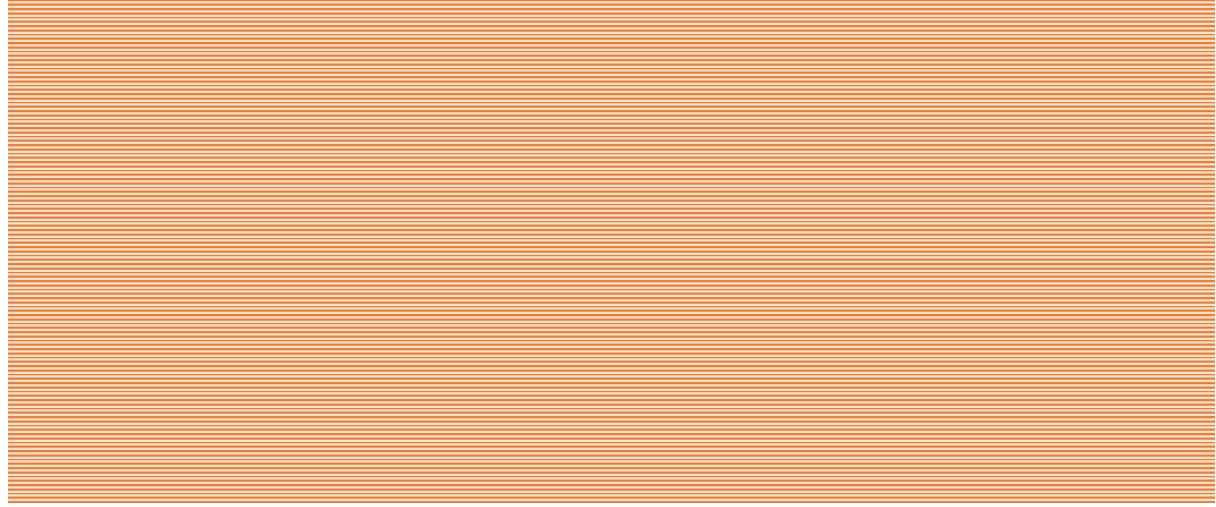

**Figure 11.** Landsat Satellite Images of the Kabul Urban Area in 2013 and 2020.

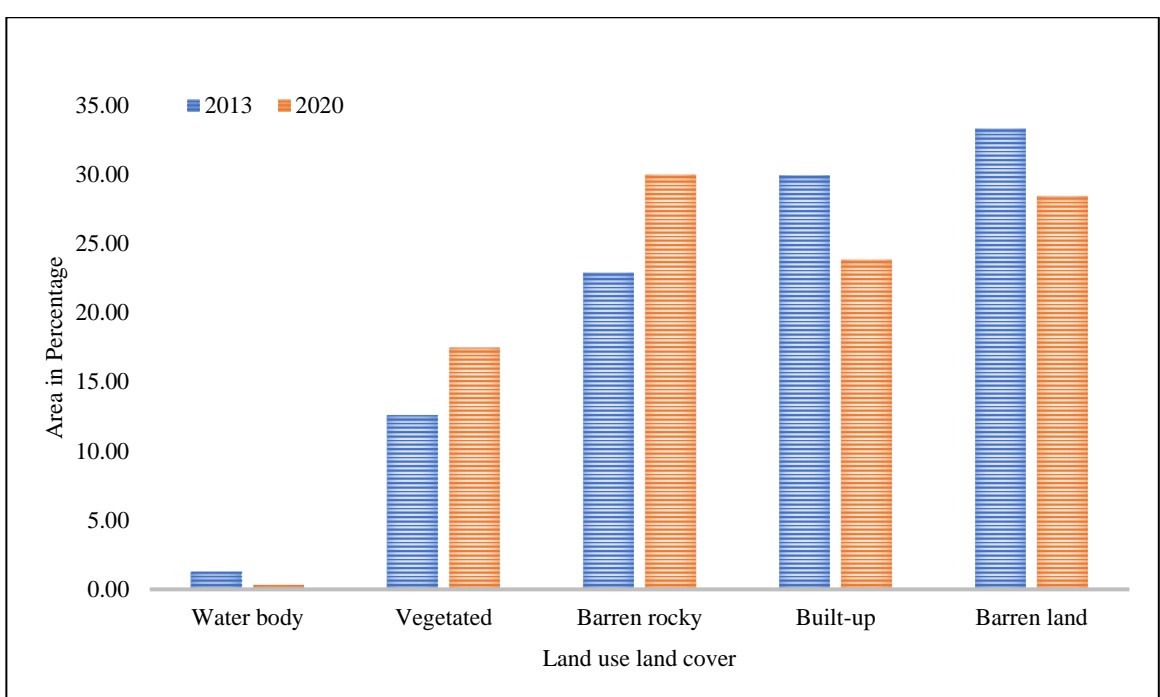

**Figure 12.** Graphical representation of land use/land cover classification during 2013–2020.

**Table 7.** Land use/land cover classification of satellite images on 15 June 2013 and 12 July 2020.

| Year | Vegetation | | Built-up | | Water Bodies | | Barren Land | | Barren Rocky | |
|------|------------|-----|----------|-----|--------------|-----|-------------|-----|--------------|-----|
| | Km² | (%) | Km² | (%) | Km² | (%) | Km² | (%) | Km² | (%) |
| 2013 | 126.7 | 12.3 | 308.0 | 29.91 | 16.5 | 1.60 | 343.0 | 33.3 | 235.8 | 22.89 |
| 2020 | 179.8 | 17.46 | 245.6 | 23.84 | 23.84 | 0.34 | 292.5 | 28.4 | 308.7 | 29.97 |

*3.6. Long-Term Land Use Land Cover Changes in Kabul during 1973–2020*

This section endeavors to scrutinize and comprehend the transformations in land use and land cover in the Kabul Urban Area (KUA) over a span of nearly five decades, specifically from 1973 to 2020. By assessing these alterations, we aim to expose the complexities and dynamism of Kabul's evolving urban landscape and illustrate the factors influencing such changes. Armed conflict and sociopolitical instability have played significant roles in these transitions and will be further explored in the subsequent analysis.

We observed a substantial increase in the built-up area from 21.39% to 26.07% between 1973 and 1983 (Figures 13 and 14). However, as echoed by Mohamed et al. [62], our study also suggests that sociopolitical disturbances, most likely due to armed conflicts, led to a decrease in the built-up area to 22.32% by 1993. The period from 1993 to 2003 was characterized by a lack of significant land use changes, likely attributable to continued internal conflict, paralleling the trends noticed by Aung [63] in conflict-stricken regions of Myanmar. An exception was a notable increase in surface water, rising from 0.5% in 1993 to 2.51% in 2003. The post-conflict decade of 2003–2013 experienced an uptick in the built-up area from 20.4% to 29.91%, suggesting a return of the population and rapid urbanization, as was similarly noted by Mhanna et al. [64] in the Syrian portion of the Orontes River Basin post-conflict. From 2013 to 2020, the built-up area decreased to 23.84%, while barren rocky terrain increased from 22.89% to 29.97%. This trend may suggest land degradation and aligns with Toro et al. [65] who argued that conflict and socioeconomic factors could drive land cover changes, impacting carbon storage and emissions also reported by Hendrix and Salehyan [66,67]. Examining these trends within the context of the ongoing armed conflict in Afghanistan offers deeper insights. Armed conflicts have profound, often detrimental

impacts on the environment and land systems, as seen in the Syrian context by Mhanna et al. [64]. For instance, the decline in the built-up area post-2013 could be a result of the adverse impacts of armed conflict, leading to infrastructural damage and reduced habitability, a trend seen in Rakhine, Myanmar [63].

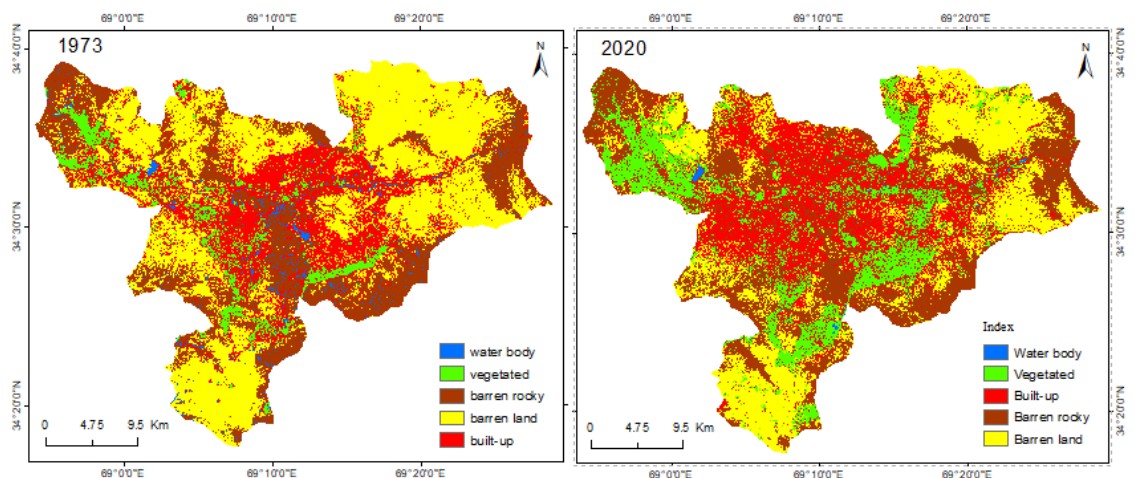

**Figure 13.** Landsat Satellite Images of the Kabul Urban Area in 1973 and 2020.

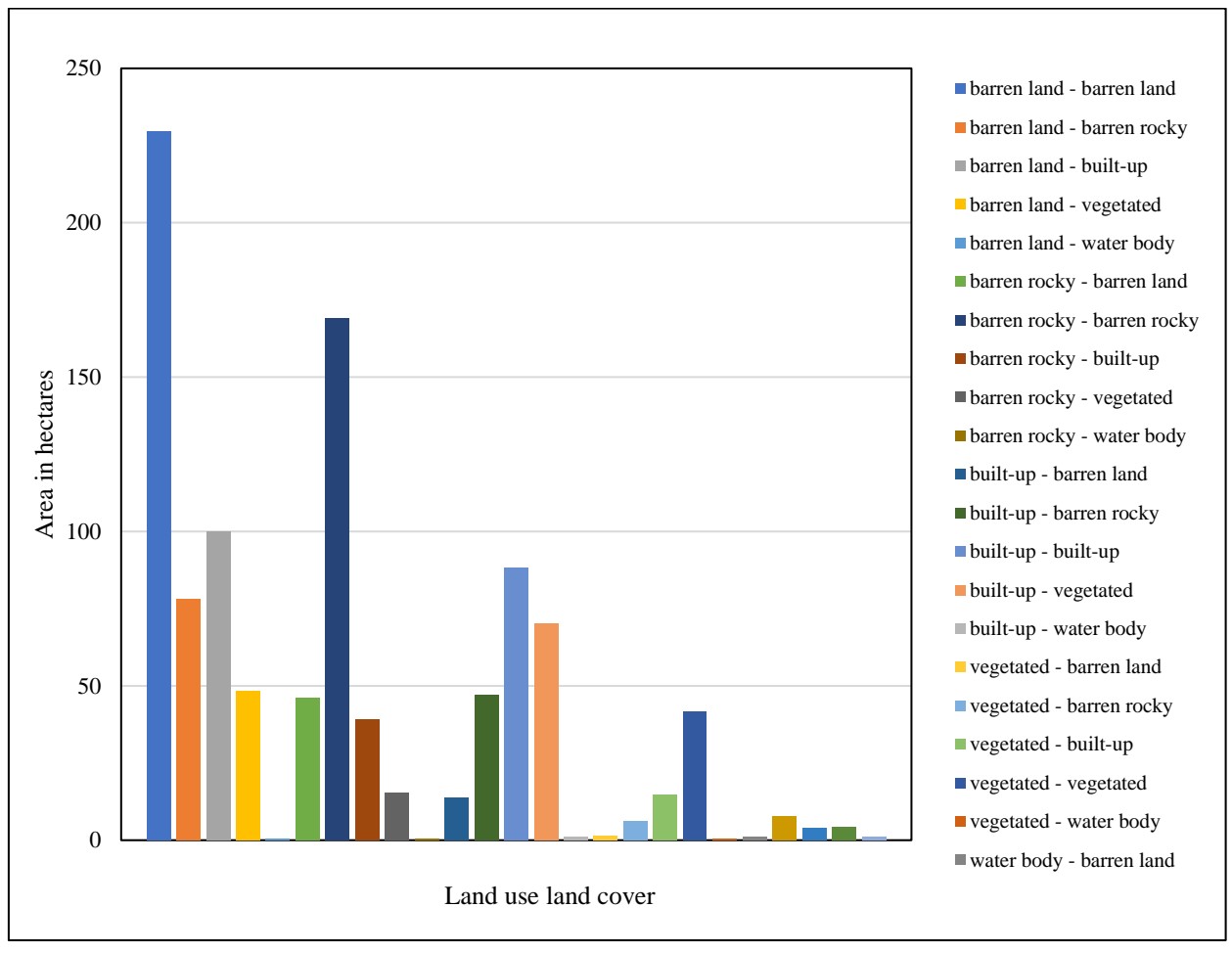

**Figure 14.** Comparison between land use/land cover change distributions during 1973–2020.

Moreover, the rise in barren, rocky terrain may signal increased land degradation due to conflict-induced population displacement, destruction in military operations, or

other conflict-related demographic changes, as suggested by Mohamed et al. [62]. Thus, armed conflict might be indirectly responsible for the altered land use, where land once cultivated or inhabited has now been abandoned and left degrade. In sum, the land use and land cover changes in the Kabul Urban Area reflect the city's dynamic response to various socioeconomic factors, including armed conflict. These transitions underscore the need for strategic land management and urban planning to balance urbanization with environmental sustainability in conflict-affected regions (Figure 15). Through understanding these processes, we can help contribute to a more sustainable and peaceful future for Kabul and other urban landscapes facing similar challenges.

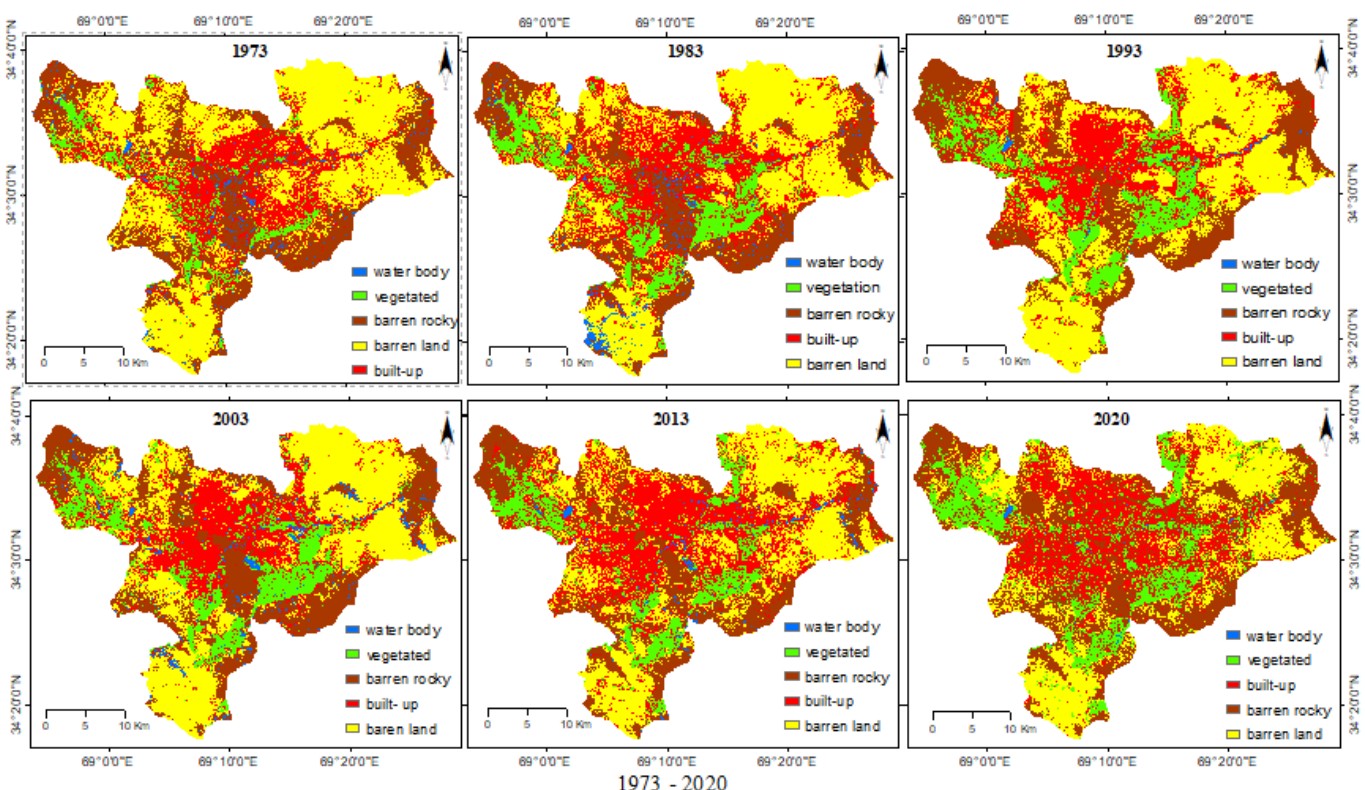

**Figure 15.** Composite LULC map during 1973–2020.

**4. Policy Recommendations**

Based on the findings of this study, the following policy recommendations are proposed:

- Given the decrease in built-up areas and the increase in barren rocky terrain, policymakers should invest in urban planning and development strategies. This could include converting suitable barren rocky terrain into built-up areas to accommodate the city's growing population.
- With the decrease in water bodies and surface water, it is evident that water management needs to be improved. Policies should be established to protect existing water bodies and to develop infrastructure that can efficiently manage surface water.
- The decrease in vegetation over the years should prompt the implementation of green initiatives. Policymakers should strive to strike a balance between urban development and environmental preservation, promoting initiatives such as green spaces and urban forests.
- Given the relatively stable vegetated area, there is potential for the development of urban agriculture. This could provide the dual benefits of increasing the city's food security and adding more green spaces to the urban landscape.

## 5. Conclusions

The present research utilizes Landsat and Sentinel satellite images to scrutinize the urban area of Kabul, specifically examining five types of land cover classes, namely water bodies, vegetation, barren land, barren rocky, and built-up areas. This analysis was performed using GIS techniques. The principal focus of the study revolves around evaluating the long-term progression of built-up areas in Kabul from 1973 to 2020. In conjunction with this, a temporal examination was carried out for the years 1973, 1983, 1993, 2003, 2013, and 2020. Over the investigated period, significant changes to the cityscape of Kabul were observed. One major transformation was noted in the barren land class between 2013 and 2020. The proportion of built-up land contracted from 29.91% in 2013 to 23.84% in 2020, while the extent of barren land also shrank from 33.3% to 28.4%. Conversely, barren rocky terrain exhibited an expansion from 22.89% in 2013 to 29.97% in 2020. Regarding water bodies, they constituted 2.51% of the total area in 2003, but their share diminished to 1.30% by 2013. Simultaneously, vegetated land use also declined from 13.61% in 2003 to 12.6% in 2013. In 1993, built-up areas comprised 22.32% of the total area, which further declined to 20.40% by 2003. This reduction in built-up areas corresponded with the mismanagement of surface water, leading to its decrease from 2.1% in 1983 to 0.5% in 1993. On the other hand, barren land experienced a rise from 35.53% in 1983 to 38.86% in 1993. The vegetated area saw a modest increase from 12.86% in 1983 to 13.98% in 1993. Looking further back, in 1973, barren land accounted for a sizable 44.35% of the area, which was reduced to 35.53% by 1983. Water bodies, which accounted for 1.76% in 1973, saw an increase to 2.1% in 1983. The proportion of barren rocky terrain, at 26.29% in 1973, saw a decrease to 23.45% by 1983. Despite these insights, the study faces some limitations, mainly its dependence on satellite imagery for land use classification. Although remote sensing technology provides a broad overview, it lacks the precision that ground-level data can furnish. The classifications in this study rely on a machine learning-based approach. As a result, they may not accurately reflect actual land use, particularly when different land use categories share similar visual characteristics. Additionally, the study does not account for potential impacts of future changes in climate or sociopolitical circumstances, which could significantly influence land use patterns. Lastly, the study's data only extend until 2020; thus, they may not represent the most current state of land use in Kabul. However, regardless of these limitations, the study does provide valuable insights into the changes in land use and land cover in Kabul over the last five decades, filling a knowledge gap due to the lack of information from local authorities. These findings are beneficial for local, regional, and urban planners in managing urban development, strategizing future urban growth, and addressing environmental concerns amid rapid urbanization.

**Author Contributions:** This research article was prepared by H.H. The concept and background theory was implemented by S.K.S., S.K. and G.M. The editing and data compiling was performed by H.H., T.A. and P.K. All authors have read and agreed to the published version of the manuscript.

**Funding:** This study received no external funding. Also, the authors declare that they have no known competing financial interests or personal relationships that could have appeared to influence the work reported in this paper.

**Institutional Review Board Statement:** Not applicable.

**Informed Consent Statement:** This research article does not involve human intervention.

**Data Availability Statement:** The datasets used in this article is free available (https://www.usgs.gov, https://www.esa.int/ accessed on 31 March 2021).

**Acknowledgments:** The authors are thankful to the USGS for providing the satellite datasets and to the reviewers of this journal for their kind reading and provision of useful suggestions. This research constitutes a segment of the M. Tech Geoinformatics dissertation undertaken by the first author, H.H., at Suresh Gyan Vihar University (SGVU) in Jaipur. The first author extends profound gratitude to Sunil Sharma and Sudhanshu of SGVU for their unwavering support and cooperation throughout the duration of the course. Additionally, we would like to express our appreciation to the three

anonymous reviewers, whose insightful and critical feedback significantly enhanced the quality of this manuscript.

**Conflicts of Interest:** The authors do not have any conflict of interest in this research paper.

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
