# Peer review of "Land Use and Land Cover Changes in Kabul, Afghanistan Focusing on the Drivers Impacting Urban Dynamics during Five Decades 1973–2020"

_2673-7418, doi:10.3390/geomatics3030024_

Round 1

Reviewer 1 Report

All technical writing is good and accurate, but ny big concern is the weakest literature review about how the 20-year warfare event affects land use land cover change in Kabul.  Although the 20-year warfare is not the focus of this paper, it inevitably affects the land use land cover pattern after 2000 or 2003. Neither did I see anything related to this 20-year shocking event in the analyses sections. I suggest adding several literature entries and also adding some relevant analyses in the later sections of the paper. In doing so, the paper will provide a unique angle on the current state of Kabul.

Other areas that should be changed for better clarity and therefore improve the significance of content, and interest to the readers are as follows - 

1. page 3 should be "Dfb - Warm Summer Continental Climate" to cap all the specific climate variable information.

2. page 3 2.1 Figure 1 - it is not referenced in the texts. Please add it.

3. page 4 2.2 -  the texts did not mention the 2020 image and its source. Please add it. Yes, it is mentioned only in Table 2.

4. page 4 line 144-5 - Should be "Figure 2", not "Figure 1."

5. pag 14 Figure 12 - are these numbers percentages or absolute values - acres or hectares?

6. page 15 - the analysis of 2003-2013 and 2013-2020 needs to be expanded with more information. Here can be a place to add some insights on how armed conflicts affect land use land cover change. There is some very nice literature on this topic out there. 

7. page 15 - need to have more, for example, how the current Afghanistan government manages the city's land specifically. In addition since Barren Land and Barren Rocks are the two classification classes, the authors should provide the definitions, i.e. NASA's, to distinguish them. 

Author Response

Response to Reviewer #1:

Comments 1: page 3 should be "Dfb - Warm Summer Continental Climate" to cap all the specific climate variable information.

Response 1: Authors are heartily thankful to the reviewers for their comments which certainly enrich the quality of the paper. The modification has been done by the authors.

Comments 2: page 3 2.1 Figure 1 - it is not referenced in the texts. Please add it.

Response 2: Authors are thankful to the reviewer for their comment. Figure 1 added in the text.

Comments 3: page 4 2.2 -  the texts did not mention the 2020 image and its source. Please add it. Yes, it is mentioned only in Table 2.

Response 3: Authors are thankful to the reviewer for their comment. And mentioned about the 2020 image in the specified section.

Comments 4: page 4 line 144-5 - Should be "Figure 2", not "Figure 1."

Response 4: Authors are thankful to the reviewer for their review and corrected the mentioned error and kept figure 2.

Comments 5: pag 14 Figure 12 - are these numbers percentages or absolute values - acres or hectares?

Response 5: Represented in hectare.

Comments 6: page 15 - the analysis of 2003-2013 and 2013-2020 needs to be expanded with more information. Here can be a place to add some insights on how armed conflicts affect land use land cover change. There is some very nice literature on this topic out there.

Response 6: Authors Thank you for your invaluable comments and suggestions on our manuscript. We appreciate the time and effort you have taken to improve the quality of our work.

The main concern was the need for expanded analysis and discussion of the Land Use/Land Cover changes in Kabul City during the periods 2003-2013 and 2013-2020, particularly in relation to the impacts of armed conflicts. We agree that this would enhance the depth and impact of our research, and we have now added more detail to these sections. We incorporated the insights gained from studies such as Mhanna et al. (2023), Toro et al. (2022), Aung (2021), Mohamed et al. (2020), and Hendrix and Salehyan (2012) to provide a broader perspective on how armed conflict can affect land use and land cover. These references helped us elaborate on how warfare, internal displacement of people, refugee movements, and related socio-economic disruptions can profoundly influence land use and land cover changes. We explored how these factors could be related to the increased built-up area between 2003 and 2013, as well as the rise in barren rocky terrain from 2013 to 2020. We believe that these additions significantly strengthen our analysis and understanding of the Land Use/Land Cover changes in Kabul City over the specified periods. We hope that our revised manuscript now meets your expectations and look forward to any further suggestions you may have.

Comments 7: page 15 - need to have more, for example, how the current Afghanistan government manages the city's land specifically. In addition since Barren Land and Barren Rocks are the two classification classes, the authors should provide the definitions, i.e. NASA's, to distinguish them.

Response 7: Authors are thankful to the reviewer for their suggestions. The authors had mentioned the definition of Barren Land and Barren Rocks in the methodology section.

Comments 8: All technical writing is good and accurate, but my big concern is the weakest literature review about how the 20-year warfare event affects land use land cover change in Kabul.  Although the 20-year warfare is not the focus of this paper, it inevitably affects the land use land cover pattern after 2000 or 2003. Neither did I see anything related to this 20-year shocking event in the analyses sections. I suggest adding several literature entries and also adding some relevant analyses in the later sections of the paper. In doing so, the paper will provide a unique angle on the current state of Kabul.

Response 8: We sincerely agree with your assessment and would like to emphasize that our study is one of the first comprehensive examinations of land use and land cover (LULC) changes in Afghanistan, especially during a period of such historical significance. This research was part of a broader investigation conducted by the first author when the nation was under a democratic government. As you have pointed out, this becomes even more critical given the recent political changes, and as the new government is in the early stages of establishing order. The present political climate indeed makes it challenging to thoroughly analyze the current state of LULC. However, our study provides a crucial baseline and a detailed overview of the preceding conditions, which could be vital for future investigations as the situation stabilizes. In light of your suggestion, we have expanded our discussion to highlight this novelty and relevance of our work in the revised manuscript. We have clarified the timeframe and conditions under which the research was conducted and explained how the findings can inform future studies and land use planning under the new government.

Reviewer 2 Report

Summary:

The study focuses on examining the gradual growth of urbanization within the captivating landscape of Kabul City, utilizing the remarkable tool of satellite imagery. The authors have skillfully incorporated a diverse range of references to bolster their research, demonstrating their commitment to conducting a thorough and all-encompassing study. However, the manuscript could benefit from improved clarity in its presentation, as well as a more concise and succinct summary of the key findings.

Major Comments:

Insufficient Clarity and Elaboration in Findings: Regrettably, the results section of the manuscript appears to be rather concise, leaving us longing for a more comprehensive depiction of the study's discoveries. The authors acknowledge their examination of the urban expansion in Kabul City, yet they refrain from presenting any precise numerical data or vivid visual representations to substantiate their observations.

Insufficient Elaboration and Analysis of Findings: Regrettably, the discussion section falls short in its ability to thoroughly analyze and interpret the results, as well as establish meaningful connections to the existing scientific literature. The authors should carefully consider the significance of their findings within the broader framework of existing research in the field of Geomatics. Additionally, it is crucial for them to elucidate the potential ramifications of their results, shedding light on how these findings may contribute to the advancement of Geomatics.

Methodological: The methodology section would benefit from additional elaboration, as it currently lacks the necessary level of detail required for replication purposes and comprehensive evaluation of the findings. The authors are encouraged to enhance the description of the satellite images employed, elaborate on the classification techniques employed, and expound upon the criteria utilized for assessing the expansion of built-up areas. The flow chart depicting the methodology lacks the necessary details to comprehensively grasp the sequential progression of the research steps. Enhancing the clarity and depth of information regarding each step, the tools employed, and the rationale behind the selected methods is imperative in order to provide a comprehensive understanding of the process.

The manuscript requires significant improvements in English language usage. There are numerous instances of grammatical errors, awkward sentence constructions, and unclear expressions that hinder comprehension.

Author Response

Comments 1: Insufficient Clarity and Elaboration in Findings: Regrettably, the results section of the manuscript appears to be rather concise, leaving us longing for a more comprehensive depiction of the study's discoveries. The authors acknowledge their examination of the urban expansion in Kabul City, yet they refrain from presenting any precise numerical data or vivid visual representations to substantiate their observations?

Response 1: Thank you for your feedback regarding the lack of precise numerical data and visual representations in our findings. In response to your comment, it's crucial to note that the research presented is one of the first of its kind, focusing on the LULC changes in Afghanistan. Our study was conducted during the time when Afghanistan was under a democratic government, and as such, it provided us with a unique opportunity to undertake this analysis. With regards to the current political climate, it is indeed challenging to gather new data and to perform a thorough analysis given the recent shift in governance. Despite this, we strive to offer a robust and accurate representation of the situation to the best of our abilities using the available data. In order to address your concerns, we plan to extend the results section by incorporating more precise numerical data that were obtained during the study period. We agree that this will enrich our study by providing a more detailed perspective of the urban expansion within Kabul City. In addition, we aim to incorporate additional visual aids, such as maps and graphs, to better illustrate the changes observed in land use and land cover over the years. We hope that these proposed changes will strengthen our manuscript by addressing your concerns about the level of detail in our findings. Once again, we thank you for your insightful feedback which serves to enhance the quality of our research.

Comments 2: Insufficient Elaboration and Analysis of Findings: Regrettably, the discussion section falls short in its ability to thoroughly analyze and interpret the results, as well as establish meaningful connections to the existing scientific literature. The authors should carefully consider the significance of their findings within the broader framework of existing research in the field of Geomatics. Additionally, it is crucial for them to elucidate the potential ramifications of their results, shedding light on how these findings may contribute to the advancement of Geomatics.

Response 2: We thank the worthy reviewer for these comments. We have improved the literature review of the discussion section. Notably, we have tried to discuss our findings with the LULC changes in the conflict zones. The added literature has added a new value to our study. We thank you for your comments.

While in other sections we added references the last section we revised it all together. The same is shown here:

3.6 Long-term Land Use Land cover changes in Kabul City during 1973 – 2020

This section endeavors to scrutinize and comprehend the transformations in land use and land cover in the Kabul Urban Area (KUA) over a span of nearly five decades, specifically from 1973 to 2020. By assessing these alterations, we aim to expose the complexities and dynamism of Kabul's evolving urban landscape and illustrate the factors influencing such changes. Armed conflict and sociopolitical instability have played significant roles in these transitions and will be further explored in the subsequent analysis.

We observed a substantial increase in the built-up area from 21.39% to 26.07% between 1973 and 1983. However, as echoed by Mohamed et al. [62], our study also suggests that sociopolitical disturbances, most likely due to armed conflicts, led to a decrease in the built-up area to 22.32% by 1993.

The period from 1993 to 2003 was characterized by a lack of significant land-use changes, likely attributable to continued internal conflict, paralleling the trends noticed by Aung [63] in conflict-stricken regions of Myanmar. An exception was a notable increase in surface water, rising from 0.5% in 1993 to 2.51% in 2003.

The post-conflict decade of 2003-2013 experienced an uptick in the built-up area from 20.4% to 29.91%, suggesting the return of population and rapid urbanization, as was similarly noted by Mhanna et al. [64] in the Syrian portion of the Orontes River Basin post-conflict.

From 2013 to 2020, the built-up area decreased to 23.84%, while barren rocky terrain increased from 22.89% to 29.97%. This trend may suggest land degradation and aligns with Toro et al. [65] who argued that conflict and socioeconomic factors could drive land cover changes, impacting carbon storage and emissions also reported by Hendrix and Salehyan [66].

Examining these trends within the context of the ongoing armed conflict in Afghanistan offers deeper insights. Armed conflicts have profound, often detrimental impacts on the environment and land systems, as seen in the Syrian context by Mhanna et al. [64]. For instance, the decline in the built-up area post-2013 could be a result of the adverse impacts of armed conflict, leading to infrastructural damage and reduced habitability, a trend seen in Rakhine, Myanmar [63].

Moreover, the rise in barren, rocky terrain may signal increased land degradation due to conflict-induced population displacement, destruction from military operations, or other conflict-related demographic changes, as suggested by Mohamed et al. [62]. Thus, armed conflict might be indirectly responsible for the altered land use, where lands once cultivated or inhabited have now been abandoned and left to degradation.

In sum, the land use and land cover changes in the Kabul Urban Area reflect the city's dynamic response to various socioeconomic factors, including armed conflict. These transitions underscore the need for strategic land management and urban planning to balance urbanization with environmental sustainability in conflict-affected regions. Through understanding these processes, we can help contribute to a more sustainable and peaceful future for Kabul and other urban landscapes facing similar challenges.

Comments 3: Methodological: The methodology section would benefit from additional elaboration, as it currently lacks the necessary level of detail required for replication purposes and comprehensive evaluation of the findings. The authors are encouraged to enhance the description of the satellite images employed, elaborate on the classification techniques employed, and expound upon the criteria utilized for assessing the expansion of built-up areas. The flow chart depicting the methodology lacks the necessary details to comprehensively grasp the sequential progression of the research steps. Enhancing the clarity and depth of information regarding each step, the tools employed, and the rationale behind the selected methods is imperative in order to provide a comprehensive understanding of the process.

Response 3: The authors are thankful for the suggestions and included the detailed steps used in the flow chart in page 4.

Comments 4: The manuscript requires significant improvements in English language usage. There are numerous instances of grammatical errors, awkward sentence constructions, and unclear expressions that hinder comprehension.

Response 4: The manuscript has been now corrected for English language errors as suggested by the reviewers. Thank you for your suggestion.

Reviewer 3 Report

The manuscript presented a multidecadal analysis of urban expansion in the city of Kabul, in Afganistan, using LUCC maps derived from remote sensing data.

Overall, the work is well designed and presented, especially in the analysis of results section.

However, the methodology could be improved. For example, it is not clear how the authors dealt with the different spatial resolutions of the different sensors used. As it is a quantitative analysis, it can result in different values. I suggest a better discussion of this point.

I also suggest the authors evaluate a post-processing step, in order to diminish the number of single pixels classified in a neighborhood. This might also influence the quantitative analysis.

In Table 1,  update the satellite for sensor MSI (Sentinel 2).

All the Figures need a better resolution.

English is good, but there are some paragraphs that need improvement, for example:

"The supervised classification method has been considered as the most popular classification method with a maximum likelihood algorithm so the supervised classification method was used for preparing Land use/land cover classified images"

"Training sets for each class  were considered around as the quality of classified images depends on validity of training sets"

"The Accuracy assessment has been performed for all the six classified images with the help of satellite images and Google Earth to verify the corresponding datasets and Kappa Coefficient was deduced"

These are difficult to understand. I suggest an overall language revision.

Author Response

Response to Reviewer #3:

Comments 1: Overall, the work is well designed and presented, especially in the analysis of results section. However, the methodology could be improved. For example, it is not clear how the authors dealt with the different spatial resolutions of the different sensors used. As it is a quantitative analysis, it can result in different values. I suggest a better discussion of this point.

I also suggest the authors evaluate a post-processing step, in order to diminish the number of single pixels classified in a neighbourhood. This might also influence the quantitative analysis.

Response 1: The authors are thankful for the suggestion. And mentioned the detailed steps in the methodology.

Comments 2: In Table 1, update the satellite for sensor MSI (Sentinel 2).

Response 2: The authors are thankful for the suggestion and updated the sensor MSI in Table 1.

Comments 3: All the Figures need a better resolution.

Response 3: The resolution of all images has been increased.

Comments 4: English is good, but there are some paragraphs that need improvement, for example:

"The supervised classification method has been considered as the most popular classification method with a maximum likelihood algorithm so the supervised classification method was used for preparing Land use/land cover classified images"

"Training sets for each class  were considered around as the quality of classified images depends on validity of training sets"

"The Accuracy assessment has been performed for all the six classified images with the help of satellite images and Google Earth to verify the corresponding datasets and Kappa Coefficient was deduced"

These are difficult to understand. I suggest an overall language revision.

Response 4: The authors are thankful for your kind suggestion and revised the language as per suggestions. Whereas the specified corrections has been done in the mentioned like as.

* The maximum likelihood algorithm was used in the supervised classification method to prepare the land use land cover classified images from 1973-2020.

* So, the maximum of 92 training sets were used to increase the quality of classified image.

* The Accuracy assessment has been performed for all the six classified images by verifying the features with the satellite images and Google Earth. The corresponding verification led to calculate the Kappa Coefficient for the output datasets